# Sialocele and Its Association with Hypercortisolism and Long-Term Glucocorticoid Treatment in Dogs: Retrospective Case–Control Study

**DOI:** 10.3390/ani14010120

**Published:** 2023-12-28

**Authors:** Jeong-Yeol Bae, Jung-Il Kim, Jin-Young Kim, Guk-Il Joung, Hong-Ju Lee, Jae-Beom Lee, Joong-Hyun Song

**Affiliations:** 1Department of Veterinary Internal Medicine, College of Veterinary Medicine, Chungnam National University, Daejeon 34134, Republic of Korea; qwe360076@gmail.com (J.-Y.B.); wjddlf2200@naver.com (J.-I.K.); qnrhrch36@naver.com (J.-Y.K.); kukil1503@gmail.com (G.-I.J.); 2Ulsan S Animal Medical Center, Ulsan 44726, Republic of Korea; lhj8596@naver.com; 3Department of Chemistry, Chungnam National University, Daejeon 34134, Republic of Korea; nanoleelab@cnu.ac.kr

**Keywords:** dog, hypercortisolism, long-term glucocorticoid treatment, salivary gland, sialocele, saliva

## Abstract

**Simple Summary:**

Sialocele-related concurrent diseases or factors in dogs are not well documented, and occasionally, dogs with sialocele have been found to have concurrent hypercortisolism (HC) or a history of long-term glucocorticoid (GC) treatment. In this retrospective case–control study, we investigated the association between HC, long-term GC treatment, and the diagnosis of sialocele in dogs. The occurrence of sialocele was higher in dogs with HC and those receiving long-term GC treatment, that is, treatment lasting for more than 5 months. This suggests that endogenous and exogenous GCs may influence sialocele formation. Therefore, we recommend regular monitoring for sialocele occurrence in dogs with HC or those receiving long-term GC treatment.

**Abstract:**

Dogs with sialocele often have concurrent hypercortisolism or are receiving long-term glucocorticoid treatment. However, their association has not been investigated. This retrospective matched case–control study investigated the association between hypercortisolism, long-term glucocorticoid treatment, and sialocele in dogs. We retrospectively reviewed the records from 1 January 2018 to 31 December 2022. Records of 19 dogs diagnosed with sialocele were investigated for hypercortisolism and long-term glucocorticoid treatment. Two age- and breed-matched controls for each sialocele dog (38 dogs) were investigated for the same concurrent diseases. Logistic regression analysis was used. The odds of sialocele in dogs with hypercortisolism were 15.56 times those of dogs without hypercortisolism (*p* = 0.02; 95% CI: 1.54–156.79). The odds of sialocele in dogs with long-term glucocorticoid treatment (median, 8 months; range, 5–13) were 7.78 times those of dogs without long-term glucocorticoid treatment (*p* = 0.03; 95% CI: 1.23–49.40). No associations were found between age, sex, body weight, and the presence of sialocele. The results indicate that sialocele was significantly associated with hypercortisolism and long-term glucocorticoid treatment in dogs. Therefore, dogs with hypercortisolism or receiving long-term glucocorticoid therapy should be screened for possible sialocele. Additionally, dogs with sialocele should be identified for concurrent hypercortisolism and prolonged glucocorticoid exposure.

## 1. Introduction

A sialocele, also known as a salivary mucocele, is a pathological condition characterized by the accumulation of saliva in a subcutaneous or submucosal area resulting from the leakage of saliva from a damaged salivary gland or duct [1,2]. The resulting cavities filled with saliva are typically lined with inflammatory connective tissue rather than the usual epithelial tissue [3,4]. Sialoceles occur more commonly in dogs than in cats and they are known to affect all breeds and any ages. The presence of concurrent diseases with sialocele and their potential associations have not yet been investigated in dogs. Our clinical observations prior to this study revealed that dogs presenting with sialocele often have underlying conditions such as hypercortisolism (HC) or have received long-term glucocorticoid (GC) treatment. Previous human studies have suggested that long-term GC administration affects the chemical characteristics of saliva, such as pH and viscosity, potentially leading to compromised salivary function [5,6,7].

Sialoceles are classified as sublingual, cervical, pharyngeal, zygomatic, or complex sialoceles based on their anatomical location [8,9]. The sublingual/mandibular gland and duct complex is the most commonly affected structure [4]. Although sialoceles are usually asymptomatic, they can cause symptoms such as oral bleeding, chewing-related trauma, respiratory distress, and dysphagia, depending on their location. Saliva can also irritate tissues, leading to inflammation, and the associated swelling can be firm and painful [9].

The diagnosis is based on the patient’s history, clinical signs, cytologic findings, and various types of diagnostic imaging such as radiography, ultrasonography, sialography, computed tomography (CT), and magnetic resonance imaging. Sialography, which involves the injection of iodinated, water-soluble contrast agent into a salivary duct, may be used to confirm the diagnosis or determine the site of origin [10,11,12,13]. While the exact cause of sialoceles remains unidentified, they often occur as a result of traumatic causes, including both surgical and nonsurgical factors, as well as inflammatory blockage or rupture of the salivary ducts. Additionally, they have reportedly been caused by factors like foreign bodies, sialoliths, and neoplasia [3,4,14,15].

This study aimed to determine whether there is an association between HC, long-term GC treatment, and the diagnosis of sialocele in dogs. We hypothesized that the odds of being diagnosed with sialocele would be higher in dogs diagnosed with HC or receiving long-term GC therapy compared to dogs without these conditions.

## 2. Materials and Methods

### 2.1. Study Design

A retrospective matched case–control study was conducted. The medical records from dogs presented to the Veterinary Teaching Hospital of Chungnam National University between 1 January 2018 and 31 December 2022 were reviewed and assessed for the presence or absence of sialocele. Dogs with sialocele were assigned to the case group, while those without sialocele were assigned to the control group. From both groups, the following information was extracted: breed, age, sex, body weight, underlying diseases, the presence of HC, and the duration and dosage of GC use. Long-term GC treatment was defined as lasting more than 5 months. A dog was considered to have HC if it had a historical diagnosis or if it presented with compatible clinical signs (e.g., polydipsia, polyuria, polyphagia, panting) and physical examination findings (e.g., hair loss, thin skin, abdominal distension) in conjunction with a positive result in either the low-dose dexamethasone suppression test or the adrenocorticotropic hormone stimulation test, or both.

### 2.2. Case Group

Nineteen dogs with definitively diagnosed sialocele were retrospectively identified from the hospital database. Sialocele was diagnosed based on history, clinical signs, and cytology findings, with additional diagnostic imaging such as radiography (MDXP-40, MEDIEN, Republic of Korea), ultrasonography (ultrasound scanner, iU22^®^, Philips, Cambridge, MA, USA), and CT (CT scanner, Alexion, Toshiba medical systems, Tochigi, Japan) used to confirm and differentiate other salivary gland diseases [10,16]. All dogs in the case group were evaluated for their underlying diseases and medications in use. Within the case group, we further categorized the dogs into subgroups: dogs with long-term GC administration, dogs with concurrent HC, and dogs with sialocele only (referred to as the remaining dogs).

### 2.3. Control Group

In total, 38 dogs were included in the control group without sialocele, and they were selected retrospectively using incidence density sampling to match 1:2 cases with controls. Additionally, each control was matched for age (±2 years) and breed within the same period as the case group [17]. A comprehensive retrospective analysis of all cases included physical examination, complete blood count (hematology analyzer, ProCyte Dx^®^, IDEXX, Portland, ME, USA), biochemistry (chemistry analyzers, Catalyst One, IDEXX, Westbrook, ME, USA), venous blood gas analysis (Stat Profile pHOx Ultra; Nova Biomedical Corporation, Waltham, MA, USA), radiography, and ultrasonography, and a review of drug usage history. The analysis focused on determining the presence of sialocele and the history of HC or long-term GC treatment. No gross findings or clinical signs, such as salivary gland enlargement and pain, were observed, and no specific abnormalities were found on imaging tests. Additionally, CT scans of the heads were conducted on 20 out of the 38 dogs for various reasons (e.g., nasal and sinus diseases, tumors). Based on the examinations, it was further confirmed that there were no abnormalities in the salivary glands.

### 2.4. Statistical Analysis

Multivariable logistic regression analyses were used to evaluate the potential association between explanatory variables (age, sex, body weight, alanine aminotransferase, alkaline phosphatase, HC, and long-term GC) and sialocele. Variables that demonstrated a univariate association with each outcome variable (*p*  ≤  0.10) were selected for inclusion in the final multivariable logistic regression models. The odds ratio and corresponding 95% confidence intervals were computed. For all statistical analyses, significance was considered with a threshold of *p* < 0.05. These complex statistical analyses were performed using SPSS software statistics (SPSS 19.0.0 for Windows, IBM, Armonk, NY, USA).

## 3. Results

### 3.1. Clinical Presentation

In total, 879 dogs were reviewed in this study. Among them, 51 had received long-term GC treatment and 21 were diagnosed with HC. Out of the 51 dogs with long-term GC treatment, 4 had sialocele, and among the 21 dogs with HC, 4 had sialocele. Additionally, among the 807 dogs without a history of long-term GC treatment or HC, 11 were diagnosed with sialocele.

This study included a total of 57 dogs, with 19 dogs in the case group and 38 dogs in the control group. The signalment of cases and controls identified during the same period is presented in Table 1. Both groups had a median age of 11 years (range, 3–17). The median body weight for the case group was 3.8 kg (range, 2–17), while for the control group, it was 5.1 kg (range, 2.4–28.1). In the case group, there were 11 female and 8 male dogs, whereas the control group consisted of 23 female and 15 male dogs. Breeds represented in the case group were Maltese (*n* = 7), Pomeranian (*n* = 4), Poodle (*n* = 2), Mixed (*n* = 2), Jindo (*n* = 2), French Bulldog (*n* = 1), and Boston Terrier (*n* = 1), while in the control group, each breed had twice the number of dogs compared to the case group. We investigated factors such as a history of trauma, foreign bodies, sialoliths, and neoplasia associated with sialocele induction in all dogs within the case group. There were no cases associated with foreign bodies, sialoliths, or neoplasia. Regarding traumatic etiology, the majority had no previous history of trauma, except for one case with an unclear history of trauma. Four dogs in the case group and two dogs in the control group received long-term GC administration. Additionally, four dogs in the case group and one dog in the control group were diagnosed with concurrent HC.

In the case group, four dogs were administered prednisolone (PDS) for >5 months (range, 5–13). Their underlying diseases were inflammatory pseudotumor (*n* = 1), caudal occipital malformation and syringomyelia (*n* = 1), necrotizing meningoencephalitis (*n* = 1), and chronic otitis (*n* = 1). PDS was prescribed at a median dosage of 2 mg/kg/day (range, 1–3). The time of identifying sialocele after starting PDS treatment was a median of 6.5 months (range, 6–10). In three of the four dogs, sialocele was found during the PDS tapering process. In the case of the remaining one dog, sialocele was identified at the time of tapering off medication after 5 months of PDS administration. Additionally, four dogs presented with concurrent HC as an underlying disease in the case group. In three of the four dogs, sialocele was diagnosed at the time of HC diagnosis. The one remaining dog was diagnosed with sialocele 9 months after being diagnosed with HC while undergoing treatment with trilostane (VETORYL^®^, Dechra, Northwich, UK) at a dosage ranging from 0.5 to 4 mg/kg twice daily. However, at the time of diagnosis of sialocele, the patient consistently exhibited clinical signs associated with HC.

The number of dogs, and type and location of sialocele within subgroups in the case group are presented in Table 2. In the case group, the mandibular gland was affected in 18 of the 19 dogs, and the cervical gland was affected in only 1 dog. Furthermore, 12 of the 19 dogs had unilateral involvement, while the remaining 7 dogs had bilateral involvement. When considering subgroups, the mandibular gland was affected in all dogs with long-term GC administration (*n* = 4) or concurrent HC (*n* = 4). Within these two subgroups, three out of four dogs in each subgroup had bilateral involvement, while the remaining one dog in each subgroup had unilateral involvement. Among the case group without long-term GC administration or concurrent HC (*n* = 11), the mandibular gland was affected in 10 of the 11 dogs, and the cervical gland was affected in only 1 dog. Furthermore, 10 dogs had unilateral sialocele, while only 1 dog had bilateral involvement.

### 3.2. Association between HC, Long-Term GC Treatment, and Sialocele

Based on univariate logistic regression analyses, we further used a forward conditional multivariable logistic regression model to analyze the relationship between variables (alkaline phosphatase, HC, and long-term GC) and sialocele. As shown in Table 3, HC and long-term GC administration were significantly associated with sialocele. The odds of sialocele in dogs with HC were 15.56 times higher those in dogs without HC (*p* = 0.02; 95% CI 1.54–156.79). In addition, the odds of sialocele in dogs with long-term GC treatment were 7.78 times higher than those in dogs without long-term GC treatment (*p* = 0.03; 95% CI 1.23–49.40). There were no statistically significant associations between age, sex, body weight, alanine aminotransferase, alkaline phosphatase, and the presence of sialocele in the multivariable model.

## 4. Discussion

In veterinary medicine, no studies have reported associations between sialocele and concurrent diseases or other factors that could increase the incidence of sialocele. In this retrospective study, the prevalence of sialocele among HC/long-term GC-treated dogs (*n* = 72) and in the complete hospital case number of dogs exclusive of HC/long-term GC-treated dogs (*n* = 807) was found to be 11.11% (8/72) and 1.36% (11/807), respectively. This supports the authors’ clinical observations. Moreover, this study observed that dogs with HC and long-term GC administration had a higher incidence of sialocele than dogs without these conditions. This suggests that both endogenous and exogenous GC may potentially affect the histological structure or physiological function of salivary glands.

The exact pathophysiology of sialocele remains unknown, but it is believed to be triggered by damage to the salivary gland or duct caused by factors such as trauma, foreign bodies, sialoliths, and neoplasia [3,4,14,15]. In this study, no individuals within the case group were associated with foreign bodies, sialoliths, or neoplasia. A history of trauma could not be identified in most cases of the case group, and they were considered idiopathic. Even in the one case with a reported history of trauma, the details were unclear and were based on the owner’s speculation. Similar to the findings in this study, a previous retrospective study in dogs reported that only 10 out of 60 dogs diagnosed with sialocele had a history of traumatic causes [3]. These studies suggest that the incidence of sialocele secondary to mechanical trauma might be less prevalent than previously known, indicating that, in addition to external factors such as trauma, internal factors may influence the incidence of sialocele.

Sialocele can occur at any age, but it is known to be more common in young dogs. In a retrospective study of 60 sialocele cases, the incidence was reported as 16.6% (10/60) in dogs under 1 year old, 11.6% (7/60) in those aged 1–2 years, and 8.3% (5/60) in those aged 2–3 years [3]. However, in our study, the age distribution of the case group was represented by a median of 11 (3–17). This is considered a limitation due to the retrospective nature of the study and the inclusion of a limited number of cases. Considering these prevalence trends and the results of this study together, further investigation is needed to determine whether concurrent HC or long-term GC treatment has a more pronounced impact on younger dogs.

Concurrent HC and long-term GC administration are associated with the systemic effects of endogenous and exogenous GC. Within the case group, among the eight dogs associated with long-term GC administration and concurrent HC, 75% (6/8) had bilateral involvement of the salivary glands with sialocele. However, among the remaining eleven dogs in the case group which were not associated with long-term GC administration or concurrent HC, only one dog (1/11, 9%) had bilateral involvement. In endocrine disorders, circulating concentrations of associated hormones are maintained at elevated levels within the body, which can impact various organs and soft tissues. In addition, clinical signs or organ changes due to the influence of these hormones can often manifest symmetrically or bilaterally. For example, in hypothyroidism and HC, cutaneous signs often manifest as bilaterally symmetric alopecia. In hypersomatotropism, increased growth hormone levels can lead to bilateral enlargement of the adrenal glands. Even in pituitary-dependent HC, excessive stimulation by adrenocorticotropic hormone secretion can result in bilateral enlargement of the adrenal glands being observed [18]. In this study, it was observed that a higher proportion of dogs influenced by endogenous or exogenous GC had bilateral involvement compared to dogs that were not associated with either. Considering their bilateral impact on parenchymatous organs, these features might be indicative of the hormonal influence of both endogenous and exogenous GC on the salivary glands.

Several previous studies in both humans and animals have proposed the relevant effects of GC administration on the salivary glands and the chemical characteristics of saliva. In embryonic mice, GC stimulates salivary gland morphogenesis [19], and in rats during the postnatal period, GC administration induces hypertrophy of salivary acinar cells [20]. Salivary secretion consists of two phases: primary acinar secretion and secondary modification by the duct system, which includes the transfer of electrolytes and water [5]. Long-term dexamethasone administration in mice results in the reduction of store-operated Ca^2+^ entry in salivary acinar cells, ultimately leading to hyposalivation [21]. In turn, this is thought to subsequently increase the accumulation of saliva. The use of systemic GC also reduces saliva pH, increases viscosity, and affects salivary electrolyte concentration as well as various enzymes. This leads to quantitative and qualitative changes in saliva, ultimately resulting in a decrease in overall salivary gland function [6,22]. Moreover, changes in the body’s cortisol levels can lead to alterations in salivary cortisol, which is utilized in diagnosing GC-related disorders in human medicine [23,24,25]. These studies support the association between systemic GC and its impact on salivary glands, substantiating salivary glands as organs influenced by GC. Furthermore, the higher incidence of sialocele observed in dogs influenced by systemic GC in this study also suggests that systemic GC could be an intrinsic factor in the development of sialocele.

Considering the impact of long-term GC administration, the acidic nature of saliva (reduced pH) may directly cause damage to the duct or make it more susceptible to injury caused by external trauma. These factors may indirectly contribute to an increased incidence of sialocele. Additionally, the increase in viscosity would facilitate the accumulation of saliva, thereby enhancing the likelihood of sialocele occurrence. Similarly, the hormonal effects of systemic GC can impact the circulation in parenchymatous organs. For instance, changes in bile composition have been reported in humans with HC and in guinea pigs receiving long-term GC administration, which may be associated with an increased incidence of gall bladder mucocele [17,26]. In dogs, further large-scale studies involving molecular biology are needed to investigate the specific dosage and duration of GC administration affecting salivary glands, as well as the characteristics of saliva, including acidity and viscosity. Additionally, more research is required to comprehensively understand the precise mechanisms underlying the occurrence of sialocele and identify other factors or risk factors that could contribute to an increased incidence of this condition.

In human medicine, numerous drugs and medical conditions are well documented for their impact on salivary physiology. The process of salivary secretion involves both the sympathetic and para-sympathetic nervous systems, with the action of the latter being particularly important. Stimulation of the parasympathetic nervous system leads to vasodilation, increasing saliva production [27]. Drugs with anti-alpha or beta-adrenergic effects can reduce the salivary flow rate. In contrast, drugs that increase the concentrations of neurotransmitters like acetylcholine and noradrenaline in the system can have the opposite effect. These drugs may directly or indirectly contribute to the development of salivary gland diseases such as sialolithiasis and sialadenitis. Additionally, systemic conditions such as diabetes mellitus, cystic fibrosis, multiple sclerosis, alcoholic liver cirrhosis, and kidney dysfunction have the potential to induce changes in salivary composition and volume. These alterations can affect the ability to protect against demineralization (dental decay) and also to effectively lubricate the soft oral tissues. Studies in human medicine suggest that these systemic diseases and medications can influence salivary gland function and oral hygiene [5,7]. While the influence of these medications and conditions on salivary gland function and oral hygiene in veterinary medicine has not been well identified, regular monitoring for oral diseases may be necessary for patients with drugs or diseases, given their known influence on oral health in humans. Additionally, the incidence of sialocele may increase in these patients when they receive GC treatment or have concurrent HC. Additional retrospective studies focusing on the occurrence of oral diseases should be conducted in patients influenced by systemic GC. Moreover, further research is needed to investigate whether conditions recognized for their impact on salivary glands in human medicine also increase the incidence of sialocele in dogs.

GCs are widely prescribed drugs in veterinary medicine. They have diverse effects on various tissues and their functions, including bone, cartilage, renal function, cardiovascular and respiratory function, gastrointestinal tract, central nervous system function, and thyroid function [18]. However, the impact on salivary glands has not been well established in veterinary medicine, but in humans, long-term GC administration is known to impair salivary gland function. This condition, known as dry mouth, is attributed to salivary gland dysfunction and can result in oral mucosal pain, dysphagia, stomatitis, difficulty wearing dentures, and an elevated susceptibility to dental caries and periodontal disease [21]. In this study, we have confirmed a correlation between endogenous and exogenous GC and the increased susceptibility to sialocele. Additionally, numerous studies support the association between systemic GC and its impact on the salivary glands. The exact cause of sialocele occurrence has not been identified, but it is commonly known to result from traumatic blockage or rupture of the duct or capsule of the salivary gland. Based on this study, we can consider systemic GC as an internal factor in the development of sialocele. It is crucial to closely monitor the occurrence of sialocele in dogs undergoing long-term GC administration and those with HC. It is necessary to inform the owner that the incidence of sialocele may increase in dogs under the influence of systemic GC, and routine investigation of mechanical trauma history related to the salivary gland and a thorough examination of salivary glands are needed. Moreover, additional sialography may be necessary, especially in patients with a history of GC administration for 5 months or more or concurrent HC [4,11,28]. Routine examinations can help detect sialocele early, enabling the monitoring of related clinical symptoms and facilitating timely intervention when necessary, allowing for treatment when needed. Additionally, diagnostic imaging can help determine the presence of other salivary gland disorders such as sialadenitis, sialadenosis, sialoliths, and neoplasia and monitor oral diseases. When sialocele is suspected, prompt surgical management is necessary, which involves the complete excision of the involved gland–duct complex [9,29,30,31].

This study has several limitations. First, as a retrospective study, the diagnosis of sialocele was not standardized across all cases. The diagnosis of sialocele primarily relied on medical record-based history, clinical signs, radiography, ultrasound, and cytology findings as the primary criteria. Additional diagnostic imaging, such as CT, was not performed in all cases. This limited the ability to accurately determine the exact location of sialocele and the presence of other concurrent salivary gland diseases. Second, this study has a small number of cases, which may lead to low statistical power and introduce potential bias. It can be challenging to establish statistically significant correlations; thus, further research involving a more substantial population of dogs is required to reinforce and substantiate these findings. Third, the criterion for defining long-term GC administration was arbitrarily set at 5 months. Studies in mice investigating the effects on saliva used a criterion of 6 weeks [21], while the human study on the impact of systemic GC chose 3 months as the criterion [6]. As there have been no studies in veterinary medicine regarding the long-term effects of GC administration on salivary glands and saliva, we set this 5-month criterion arbitrarily. Further basic research in veterinary medicine is necessary to explore the influence of GC on salivary glands and determine the specific duration of GC administration that impacts salivary glands. This can provide valuable insights, particularly when considering the appropriate duration of GC use, especially for dogs susceptible to sialocele development. Fourth, since there have been no studies in veterinary medicine on drugs or conditions that affect the salivary glands, our study may not have completely excluded the influence of other potential factors when evaluating the association between long-term GC use or concurrent HC and the condition of the salivary glands.

## 5. Conclusions

The study findings indicate that dogs with HC and long-term GC treatment have significantly higher odds of developing sialocele than those without these conditions. Furthermore, the contribution of trauma as a causative factor for sialocele was relatively low. We also observed a higher proportion of sialoceles with bilateral involvement in individuals influenced by systemic GC, as in some endocrine disorders. Based on the evidence from numerous studies and the findings from our study, it is evident that salivary glands are tissues influenced by systemic GC. This suggests that both endogenous and exogenous GC could be an internal factor contributing to the occurrence of sialocele. Therefore, we recommend considering the possibility of sialocele occurrence in dogs with HC or those under long-term GC therapy. Regular monitoring through physical and imaging examinations is recommended to ensure early detection and timely intervention. Additionally, dogs diagnosed with sialocele should be identified for concurrent HC if clinically suspected, as well as prolonged GC exposure. Furthermore, early detection and treatment of sialoceles in dogs with HC or receiving long-term GC therapy are crucial to prevent potential complications and improve overall patient outcomes.

## Figures and Tables

**Table 1 animals-14-00120-t001:** Signalment of dogs with sialocele and control dogs (age- and breed-matched) without sialocele.

	Dogs with Sialocele (*n* = 19)	Control Dogs (*n* = 38)
Age in years—median (range)	11 (3–17)	11 (3–17)
Bodyweight in kilograms (range)	3.8 (2–17)	5.1 (2.4–28.1)
Sex		
Female	11 (58%)	23 (61%)
Male	8 (42%)	15 (39%)
Breed		
Maltese	7 (37%)	14 (37%)
Pomeranian	4 (21%)	8 (21%)
Poodle	2 (11%)	4 (11%)
Mixed	2 (11%)	4 (11%)
Jindo	2 (11%)	4 (11%)
French Bulldog	1 (5%)	2 (5%)
Boston Terrier	1 (5%)	2 (5%)

**Table 2 animals-14-00120-t002:** Number of dogs, and type and location of sialocele by subgroup within the case group.

	Dogs with Long-Term GC Administration	Dogs with Concurrent HC	Dogs with Sialocele Only
Number of dogs	4	4	11
Type of sialocele (number)	Mandibular (4)	Mandibular (4)	Mandibular (10) Cervical (1)
Location (number)	Unilateral (1) Bilateral (3)	Unilateral (1) Bilateral (3)	Unilateral (10) Bilateral (1)

HC-hypercortisolism; GC-glucocorticoid.

**Table 3 animals-14-00120-t003:** Odds ratio, *p* value, and 95% confidence intervals (CI) from multivariable logistic regression analyses identifying associations between hypercortisolism (HC), long-term glucocorticoid (GC) treatment, and diagnosis of sialocele in dogs.

Variables	Odds Ratio	*p* Value	95% CI
HC	15.56	0.02	1.54–156.79
Long-term GC administration	7.78	0.03	1.23–49.40

## Data Availability

The data presented in this study are available on request from the corresponding author. The data are not publicly available due to privacy policy of the institute.

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
