# Peer review of "Sialocele and Its Association with Hypercortisolism and Long-Term Glucocorticoid Treatment in Dogs: Retrospective Case–Control Study"

_animals, 2023, doi:10.3390/ani14010120_

Round 1
Reviewer 1 Report
Comments and Suggestions for Authors
This manuscript is well-written and has a good standing for publication. However, it would have been more appropriate if younger dogs (1-3 years) were also included as sialocele is more prevalent in younger dogs.
It would have been advantageous for the manuscript if some mechanistic points regarding HAC and GC therapy were included. This warrants further investigation.
Reviewer 2 Report
Comments and Suggestions for Authors
Thank you for the manuscript and for communicating the results and perspectives of your study.
The study has scientific value as it communicates a hitherto probably overlooked sequela to hyperadrenocorticism/long-term glucocorticosteroid treatment.
The Introduction is a bit long, but it combines to the Discussion. If possible, a reduction/condensation of the text can be suggested. The reason for conducting the study (i.e., the authors' clinical observations) is revealed in the end of the Introduction. I suggest to considering placing this earlier in the Introduction.
The Materials and Methods are relevant to the problem investigated. I suggest including the calculation of the prevalence of sialocele among HAC/Steroid treated dogs and in the complete hospital case number of dogs exclusive HAS/steroid treated dogs for the period investigated to substantiate the authors' clinical observations. In the control group, it is noted that 20 of 38 dogs had CT scans performed. Were these CT scans done to specifically search for sialocele (the wording suggests that the CT scans were done specifically to rule-in/rule out sialocele). I suggest including a reference to the statistical tests applied as a service to the readers. Were a 1:2 ratio between cases and controls arbitrarily chosen or can this be backed up with a reference to a publication?
Results: Clearly presented. In Table 2, include a footnote explaining GC and HAC. Does the difference in location (bilateral /unilateral) differ significantly between dogs with HAC/GC and dogs without HAC/GC? If so, this could be included in the Discussion (in the sections describing and discussing location). Table 3: Consider reducing the number of significant digits in the OR column (in the text only 2 significant digits are used). OR should be explained (e.g in the table legend Odds ratio (OR) ).
Discussion: Perhaps a bit long but I cannot point to areas or sections that can be omitted.
Conclusions: Are grounded in the results of the study.
Reviewer 3 Report
Comments and Suggestions for Authors
Thank you for the opportunity to review this manuscript. In fact, the last three canine patients with sialocele that I saw were patients with Cushing´s syndrome; however, since this is the main disease we attend at our endocrine service, I did not realize that an eventual relationship could exist. This is a nice case-control study and brings new insights regarding Cushing´s possible complications, as well as a potential uncommon clinical sign. Below are some comments I would like to see addressed.
Title and the entire manuscript – I would strongly suggest “hyperadrenocorticism/HAC” be modified by “hypercortisolism/HC” as suggested by the ALIVE Project from the European Society of Veterinary Endocrinology. More definitions can be found on their website. https://www.esve.org/alive/intro.aspx
Study Design - It is not clear why the authors started the MMs section describing that a retrospective evaluation was conducted searching for patients with HC or GC exposure against patients without HC or with no long-term history of GC exposure. This description, in the way it is, may sound like a bias. In my opinion, the right way to design this study would be to first search for dogs with sialocele, and later look for matched controls without documented sialocele. After this, a comparison between HC occurrence or long-term GC exposure between dogs with or without sialocele could generate an Odds Ratio more properly.
Case group – how were those dogs identified? Retrospectively within the Hospital database?
Control group – how were the control dogs selected? There was any randomization in this selection? It is not clear if the control dogs were submitted to a clinical evaluation (transversal approach) or if the exams described were done in the past (retrospective approach). Please explain these details in the text.
Statistical analysis – typically a multivariate model is built after evaluation of serial univariate analysis, and not the opposite as described. Please clarify which variables were included in the univariate analysis.
Results
Lines 139-143 – The text here is hard to understand. Please review. The way is written gives the idea that all cases in the control group had factors such as trauma, foreign bodies, sialoliths, and neoplasia identified. However, the next lines do not make much sense if the first assumption is true. Please review the information.
Line 158 – please state if the dog that developed sialocele months after HC diagnosis was being treated, and if so, please describe the therapeutical modality chosen, drug name and dose, as well as a tentative estimation of the level of clinical control based in clinical signs, clinical pathology, and hormone tests.
Table 3 – If a multivariate model was built based on variables that reach significance, were all variables initially assessed significant, and some in the multivariate model lost significance? Table 3 results look like isolated univariate analysis. This aspect reinforces that the statistical methods described should be reviewed.
Discussion
Line 200 – The fact the authors didn´t see theoretical traumatic cases in this small study cannot allow an assumption that traumatic sialoceles are less common than previously reported.
Line 216 – what causes bilateral adrenal enlargement in dogs with Pituitary-dependent HC is ACTH not CRH. Also, the entire paragraph lacks references to Embase the discussion.
Line 240 – “that systemic GC could be an intrinsic factor” instead of “that systemic GC is an intrinsic factor”
Line 250 – “In dogs, further large-scale studies involving molecular biology and statistical analysis are needed to investigate the specific…” Statistical analysis is an indissociable part of any “large-scale” study.
Line 331 – “was” relatively low
Comments on the Quality of English LanguageOverall good English quality and easy to read and understand. However, eventually, some language barrier could be the cause of the doubt regarding statistical analysis.
Round 2
Reviewer 3 Report
Comments and Suggestions for Authors
Thank you for considering all suggestions to improve the manuscript quality and for bringing this "new" Cushing´s syndrome complication into light. All aspects pointed out were adequately clarified or correct. Congratulations for the work done.